# LogicInference: A New Dataset for Teaching Logical Inference to seq2seq Models

**Santiago Ontañón, Joshua Ainslie, Vaclav Cvicek & Zachary Fisher**
Google Research
Mountain View, CA 94043, USA
{santiontanon,jainslie,vcvicek,zachfisher}@google.com

## Abstract

Machine learning models such as Transformers or LSTMs struggle with tasks that are compositional in nature such as those involving reasoning/inference. Although many datasets exist to evaluate compositional generalization, when it comes to evaluating inference abilities, options are more limited. This paper presents LogicInference, a new dataset to evaluate the ability of models to perform logical inference. The dataset focuses on inference using propositional logic and a small subset of first-order logic, represented both in semi-formal logical notation, as well as in natural language. We also report initial results using a collection of machine learning models to establish an initial baseline in this dataset.

## 1 Introduction

It is well known that machine learning models such as Transformers (Vaswani et al., 2017) or LSTMs (Hochreiter & Schmidhuber, 1997) struggle with tasks that are compositional in nature (Liška et al., 2018; Hupkes et al., 2020; Keysers et al., 2019; Ontanón et al., 2021) such as those involving reasoning/inference. Many datasets have been proposed to show this effect such as SCAN (Lake & Baroni, 2018) or PCFG (Hupkes et al., 2020) among many others. However, when it comes to evaluating inference, options are more limited. This paper presents LogicInference[1], a new dataset to evaluate the ability of models to perform logical inference. The dataset focuses on inference using propositional logic and a small subset of first-order logic, represented both in semi-formal logical notation, and in natural language. LogicInference has two main long-term goals: (1) to evaluate the ability of models to perform logical inference, and the degree to which inference chains are real or hallucinated, and (2) to assess whether learning logical inference abilities in the abstract (e.g., getting better in this dataset) would then transfer to other real-world tasks.

Some datasets already exist to assess inference in machine learning models. LogicInference is designed to complement those. For example, natural language entailment datasets, such as SNLI (Bowman et al., 2015), involve only single step inferences, as the model only needs to establish the relation between two pieces of text (entails/contradicts/unrelated), but it is not asked to present the reasoning chain (although an extension with simple explanations was recently proposed in Camburu et al. (2018)). CLUTRR (Sinha et al., 2019) asks models to reason about family relation (given a story in natural language with a collection of characters, where some of their family relations are mentioned, the model needs to infer a family relation that was not mentioned). The main drawback of this dataset is that the types of reasoning it covers are limited to reason about family trees/graphs. Another dataset is EntailmentBank (Dalvi et al., 2021), which asks models to construct an entailment tree that supports a hypothesis given some text/corpus. While this dataset is a great resource for evaluation, its limited size (1840 instances in total) limits the types of experiments that can be performed on it. Also related are datasets designed to evaluate mathematical reasoning abilities. For example, the *mathematics* (Saxton et al., 2019) dataset is a relatively large dataset (2m examples) concerning math problems. One difference with our dataset is that the mathematics dataset only contains inputs and final answers, but no reasoning chain that the model has to produce. The GSM8k dataset (Cobbe et al., 2021) is a similar dataset, but that contains reasoning

---

[1] https://github.com/google-research/google-research/tree/master/logic_inference_dataset.

chains for about 8k simpler math word problems, and the Math32k dataset (Wang et al., 2017) is a middle ground, where there is no reasoning chain, but the equation used to produce the final answer is provided as part of the dataset.

The main features of LOGICINFERENCE are:

- It covers propositional logic and a subset of first-order logic[2];
- The model needs to learn a variety of tasks such as translate language to logic, one-step or multi-step inferences, and doing so in semi-formal logic notation or natural language;
- The model is asked for the step-by-step inference chains used to reach the conclusions;
- The dataset includes corner cases, such as contradictory premises, questions that do not follow from the premises, or that are obvious (answer already stated in the premise), so that the model learns to handle these cases and learns not to hallucinate inference chains when asked to infer something that cannot be inferred or that is obvious;
- The dataset contains several data splits and configurations for better evaluation.

Moreover, we acknowledge the dataset has also several important limitations, which we plan to address in the future. First, only a subset of first-order logic is covered, and only symbolic reasoning is involved (no numerical or mathematical reasoning). But second, and most important, the main limitation of the dataset is that the natural language it contains is automatically generated, and hence it does not contain the expected variety and complexity found in natural language written by humans.

In the remainder of this paper, we present the dataset in Section 2, a brief description of how it is generated in Section 3, and finally Section 4 presents the performance of a collection of machine learning models to establish baseline performance for future reference.

## 2 THE LOGICINFERENCE DATASET

LOGICINFERENCE is a sequence-to-sequence dataset, where both the input and output are strings of text. The input is a question, and the model needs to provide the answer, including any reasoning chain used to generate it. The model may be asked to provide the name of the inference rule used in each step. Examples in the dataset can belong to one of 5 different types: 1 (language to logic), 2a (one step inference), 2b (one step inference in natural language), 3a (inference chains) and 3b (inference chains in natural language) (examples in Table 1). The main motivation for having both 2a/3a and 2b/3b is that hopefully the model can learn the general principles of inference from both forms of examples, and hence generalize better when confronted with real world tasks after training in this dataset. Appendix B shows more examples of the dataset, illustrating interesting cases.

**Splits.** The dataset is synthetically generated and the generating script can generate a dataset of arbitrary size. Moreover, we provide support for three different data splits: IID, OOD, and length split. In the IID split, data is randomly split between training and test (we expect models to do well in this split). In the OOD split, data is split such that the model sees different reasoning chains during training and during testing, and hence, it should be a harder generalization challenge. Specifically, it targets the ability of the model to exhibit *systematicity* (Hupkes et al., 2020). Finally, the length split has the model training on shorter examples, and then test on longer examples. This type of length split targets the ability of the model to exhibit *productivity* (Hupkes et al., 2020), and is known to be hard for machine learning models (Lake & Baroni, 2018).

**Answer Position.** Finally, notice that in the examples in Table 1 the model is asked to provide the final answer of problem types 3a/3b in the very first token of the output. This requires the model to assess whether or not an inference chain exists to prove the target clause, with a single forward pass in the case of seq2seq models. One hypothesis we wanted to evaluate is whether giving the model the chance of performing more computation for harder problems (by asking it to generate the inference chain first, and only once the chain is generated, answer yes or not) would allow the model to generalize better. Hence, two versions of the dataset exist, LOGICINFERENCE$_b$ (which has the answer at the **b**eginning) and LOGICINFERENCE$_e$ (which has the answer at the **e**nd).

---

[2]A larger subset of first-order logic (FOL) will be included in future revisions of the dataset. Only a small subset was added in this dataset, as the inference mechanisms used to generate it were designed for propositional logic, and we added the small subet of FOL that our code could handle without problems.

| Type | Example |
|------|---------|
| 1 | **Input:** Translate the following inference to logic notation: If James were rich, then Susan is playing squash. James is rich. Therefore Susan is playing squash.
**Output:** p -> q. p. Therefore q. |
| 2a | **Input:** What can be inferred from the following premises in a single inference step (ignoring inferences that add new predicates or constants)? Name the inference rule being used: forall x_2: Q(x_2). exists x_2: P_2(x_2) -> Q_2(x_2). forall x_2: P_2(x_2).
**Output:** exists x_2: Q_2(x_2) can be inferred via the existential modus ponens rule. forall x_2: Q(x_2) and P_2(x_2) can be inferred via the universal conjunction rule. forall x_2: P_2(x_2) and Q(x_2) can be inferred via the universal conjunction rule. |
| 2b | **Input:** What can be inferred from the following premises in a single inference step (ignoring inferences that add new predicates or constants)? David is working if and only if it is cloudy. David is not working.
**Output:** If David works, then it is cloudy. If it is cloudy, then David is working. It is not cloudy. |
| 3a | **Input:** Consider the following premises. p <-> p_2. p_2 -> ~q_2. q_2. Can we infer p from them? If possible, name the inference rules being used at each step.
**Output:** Yes, via the following inference chain. From p_2 -> ~q_2, q_2 we can infer ~p_2 via modus tollens. Finally, from p <-> p_2, ~p_2 we can infer ~p via biconditional elimination. |
| 3b | **Input:** Consider the following premises. There is at least one x for which if x were an astronaut, then x is climbing a mountain. For all x, if x climbs a mountain, then x is an astronaut. Can we infer the following from them? If we can, name the inference rule being used: There is at least one x for which if x were an astronaut, then x is climbing a mountain.
**Output:** Yes, that is one of the premises. |

Table 1: Sample training examples from the LOGICINFERENCE dataset.

## 3 DATASET GENERATION

In order to generate LOGICINFERENCE, the following procedure was used:

1. Let $\mathbb{R}$ be a set of inference rules $\mathbb{R}$, such as *modus ponens*, *modus tollens*, etc. Rules include propositional logic (17 rules) and rules with universal and existential quantifiers (49 rules) (see Appendix A for examples and details).

2. Given the rules in $\mathbb{R}$, we generate a set of *inference problems* $\mathbb{P}$, where an inference problem $p = (P, I, C, U)$ is defined by a set of premises $p.P$, a set of potential inference chains $p.I$ and $p.C$ ($p.I$ containing inference chains to prove some statements, and $p.C$ to disprove some statements), a set of unrelated clauses $p.U$ (clauses that cannot be proven or disproven from the premises). This is done by starting from an inference rule, and randomly choosing other rules that could be used to infer any of the current premises, creating arbitrarily long inference chains/trees. We control for contradictions introduced by this process, but allow some in the dataset to test the model's ability to detect them.

3. For each inference problem $p$, we generate a set of *renaming variations* by renaming the propositions, constants and variables appearing in them (to prevent models memorizing patterns associated with variable names). For example, $p \rightarrow q$, could be renamed to $r \rightarrow p_2$. This results in an enlarged set of problems $\mathbb{P}_v$.

4. To generate a training example, we pick a variation $p \in \mathbb{P}_v$ at random, and then we stochastically pick one of 5 the different problem types to generate an example:

Type 1: **language to logic**: given a natural language representation of the premises and a potential inference, the model is asked to translate it to a more formal logical notation.

| | From Scratch (50k) | | | Pre-trained (20k) | | |
|---|---|---|---|---|---|---|
| LOGICINFERENCE$_b$ | small | base | large | small | base | large |
| IID | **0.890** | 0.872 | 0.793 | 0.853 | 0.821 | 0.886 |
| OOD | 0.788 | 0.750 | 0.640 | 0.774 | 0.763 | **0.789** |
| Length | 0.335 | 0.257 | 0.187 | 0.384 | 0.370 | **0.459** |
| LOGICINFERENCE$_e$ | small | base | large | small | base | large |
| IID | 0.888 | 0.859 | 0.799 | 0.819 | 0.802 | **0.905** |
| OOD | 0.788 | 0.779 | 0.660 | 0.785 | 0.777 | **0.814** |
| Length | 0.312 | 0.252 | 0.192 | 0.371 | 0.389 | **0.457** |

Table 2: Sequence-level accuracy of T5.1.1 models on the different dataset splits.

Type 2a: **one step inference**: given a set of premises, the model is asked to predict all the possible one step inferences that can be done from them.

Type 2b: **one step inference in natural language**: same as 2a, but the input and output are in natural language, rather than in logical notation.

Type 3a: **inference chains**: given a set of premises and a potential inference, the model is asked whether we can prove the inference or not from the premises, and to provide the inference chain in either case.

Type 3b: **inference chains in natural language**: same as 3a, but using natural language, rather than formal notation.

5. The *IID split* dataset is generated by trying to generate up to $n$ (n = 200k in our experiments) examples in this way and filtering for duplicates, and then splitting them randomly 90% / 10% between train / test.

6. The *OOD split* is generated by randomly splitting the set $\mathbb{P}$ into two subsets (one for training, one for testing), and then continuing the process of generation separately for each of those subsets. In this way, we can ensure that, given inference problem, all the examples derived from it are either in the training set or they are all in the test set. This creates a harder generalization challenge, as problems in the test set will have reasoning patterns (sequence of rule applications) not seen during training.

7. Finally, the *length split* is generated by splitting the examples based on the number of premises their corresponding inference problems have: less or equal to $k$ (= 4 in our experiments) premises for training, and more than $k$ for testing. With the default settings, the dataset has 118931 instances for training and 13215 for testing in the IID split, 118822 / 13266 in the OOD split, and 114790 / 17356 in the length split (See Appendix D for more detailed statistics).

**Considerations,** In problems of type 3a and 3b, the potential inference may be selected from the unrelated set (e.g., `Consider the following premises. p. p -> q. Can we infer r?`). This is to prevent the model from making up answers when users ask questions that cannot be answered from the premises. With some probability, problems contain contradictions in the premises (either direct contradictions, or their inferences reach a contradiction). This can happen in problems of types 2a, 2b, 3a and 3b. In this case, the model is supposed to identify this, and respond accordingly (anything can be inferred from a contradiction). Some inference rules can produce an infinite number of inferences, e.g.: if we know $p$, we also know $p \lor q$, $p \lor r$, $p \lor s$, etc. In problems of types 2a and 2b, the model is explicitly told not to introduce new variables or predicates, to prevent this. Finally, sometimes, the potential inference is directly present in the premises. Again these problems are introduced to force the model to detect this and respond accordingly.

## 4 BASELINES

In order to establish some baseline performance in this dataset, we evaluated the performance of a collection of T5 (Raffel et al., 2019) models. Specifically, we evaluated the *small*, *base* and *large* configurations of T5.1.1, which have 77m, 248m and 783m parameters respectively. Moreover, as previous work has shown that pre-training significantly helps in compositional generalization (Furrer et al., 2020), we evaluated both finetuning the public pre-trained checkpoints (for 20k steps), as well as training from scratch (for 50k steps). Although performance was still increasing at the end of

training, it was starting to taper off, as shown in the training curves in Appendix E. We used batch size 128, constant learning rate of 0.001, the AdaFactor (Shazeer & Stern, 2018) optimizer, and sequence lengths of 256 tokens for inputs, and 512 for targets. Table 2 show the performance of these models evaluated using *sequence level accuracy* (percentage of times they predicted the exact ground truth output sequence).

The results show that pre-training (which was known to help compositional generalization) significantly helps in the length split, where the pre-trained models significantly outperforming the non-pre-trained versions. Pre-training also seems to help the large model in all three splits. Interestingly, the small and base models do not seem to benefit from pre-training for the IID or OOD splits. The OOD split (measuring systematicity) was harder than the IID set as expected, but the length split (measuring productivity) was even harder for models. Surprisingly, we did not see any clear difference in between LOGICINFERENCE$_e$ and LOGICINFERENCE$_b$ (the large model achieves higher performance in IID/OOD with the LOGICINFERENCE$_e$ setup, but the other results are mixed). Moreover, detailed analysis shows that trained-from-scratch models struggle in problems of type 1 in the length split, while this is not as pronounced in the pre-trained models (more detailed results in Appendix E). Finally, when training from scratch, we saw the large models perform worse than the smaller models, which might be due to overfitting.

## 5  CONCLUSIONS AND FUTURE WORK

In this paper we presented LOGICINFERENCE, a new dataset designed to evaluate the inference abilities of seq2seq models, and to investigate if models trained in this dataset transfer inference abilities to real world natural language tasks (which is part of our future work). We evaluated a collection of T5.1.1 models on our dataset in order to establish an initial baseline performance as a reference for future work. As part of our future work, we would like to improve the dataset by including a larger subset of first order logic, and improve the natural language in the dataset by having humans rephrase a subset of the automatically generated inputs/output pairs in the dataset.

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

## A  APPENDIX: INFERENCE RULE DEFINITION

There are 17 propositional rules and 2 base quantified rules defined in the dataset generation script. Each rule is defined as a 5-tuple: $(P, I, C, U, name)$, where $P, I, C,$ and $U$ have the same meaning as for the inference problem definition above (premises, inferences, contradictions, unrelated), and *name* is the name of the inference rule. For example, *modus ponens* is defined as follows:

$$\begin{pmatrix} P & = & \{p \rightarrow q \ , \ p\} \\ I & = & \{q\} \\ C & = & \{\neg q\} \\ U & = & \{r, \ \neg r\} \\ name & = & \text{"modus ponens"} \end{pmatrix}$$

which, in the Python definition looks as follows:

```
([["->", ["p"], ["q"]], ["p"]],
 [["q"]],
 [["~", ["q"]]],
 [["r"], ["~", ["r"]]],
 ["p", "q", "r"],
 "modus ponens")
```

Notice that in the Python version there is an additional field (right before the name) that contains a list of all the atomic propositions involved in the rules. This is included for convenience, to prevent having to search them every time.

In addition to the 17 propositional rules, there are 2 quantified rules (*universal instantiation*, and *existential generalization*). Universal instantiation looks as follows:

```
([["forall", "var x", ["P", "var x"]]],
 [["P", "a"]],
 [["~", ["P", "a"]]],
 [["Q", "a"], ["~", ["Q", "a"]]],
 [["P", "var x"], ["P", "a"], ["Q", "a"]],
 "universal instantiation")
```

Finally, from each of the 17 propositional rules, we generate one universally quantified rule automatically by turning each predicate $p$ into $P(x)$ and adding a universal quantifier. We also generate several existentially quantified rules by universally quantifying all premises but one, and existentially quantifying one of them. Then the conclusions are existentially quantified. This generates 47 additional rules, resulting in 17 + 2 + 47 = 66 total rules used for generation of inference problems, which form the set $\mathbb{R}$.

## B  APPENDIX: MORE DATASET EXAMPLES

We include here a few more examples from the dataset, to illustrate some of its features.

### B.1  LOGICINFERENCE$_b$ VS LOGICINFERENCE$_e$

Here is the same training example for both versions of the dataset. This is how it looks in LOGICINFERENCE$_b$ (with the answer at the beginning):

**Input:** Consider the following premises.  forall x_3:  Q_1(x_3).  forall x_3: ~P_3(x_3).  forall x_3:  P_3(x_3) or Q_3(x_3).  Can we infer exists x_3:  ~Q_3(x_3) from them?

**Output:**  No, we can see why via the following inference chain.  From forall x_3: P_3(x_3) or Q_3(x_3), forall x_3:  ~P_3(x_3) we can infer forall x_3:  Q_3(x_3).  From forall x_3:  Q_3(x_3), forall x_3:  Q_1(x_3) we can infer forall x_3:  Q_3(x_3) and

```
Q_1(x_3).  Finally, from forall x_3:  Q_3(x_3) and Q_1(x_3) we can infer forall x_3:
Q_3(x_3), which contradicts exists x_3:  ~Q_3(x_3).
```

This is how it looks in LogicInference$_e$ (with the answer at the end):

**Input:** `Consider the following premises.  forall x_3:  Q_1(x_3).  forall x_3:`
`~P_3(x_3).  forall x_3:  P_3(x_3) or Q_3(x_3).  Can we infer exists x_3:  ~Q_3(x_3)`
`from them?`

**Output:** `From forall x_3:  P_3(x_3) or Q_3(x_3), forall x_3:  ~P_3(x_3) we can infer`
`forall x_3:  Q_3(x_3).  From forall x_3:  Q_3(x_3), forall x_3:  Q_1(x_3) we can`
`infer forall x_3:  Q_3(x_3) and Q_1(x_3).  Finally, from forall x_3:  Q_3(x_3) and`
`Q_1(x_3) we can infer forall x_3:  Q_3(x_3), which contradicts exists x_3:  ~Q_3(x_3).`
`Therefore, the answer is no.`

Notice that the input is the same, and the only change in the output is that the answer ("No") appears either in the first token, or in the last token of the output. The main goal of doing this variation is as follows: if the answer is in the first token, the model is forced to already predict (with a single forward pass of the Transformer encoder in our experiments) the final answer of the problem, regardless of how long is the inference chain required to assess this answer. If a human were to solve this problem, she would probably use some scrap paper to write down some inferences before reaching the final answer. We hoped to use the inference chain as this "scrap paper", before reaching the final decision (although our experiments show that this had only a small effect in performance).

### B.2    Contradictions, Unrelated and Obvious Answers

As mentioned above, some examples do not require any inference chain, but contain unrelated or obvious questions, or even contradictory premises. We added these examples (in a small proportion) to teach the model that it is not always the case that it has to look for an inference chain, and some times the question is either obvious or does not make sense. Hopefully, this contributes to minimize hallucination when training on this dataset and trying to transfer logical inference skills to another task.

This is an example where the answer is obvious, as it is stated as one of the premises:

**Input:** `Consider the following premises.  There is at least one x_1 for which x_1 is`
`a lawyer.  For all x_1, x_1 will go running.  Can we infer the following from them?`
`If we can, name the inference rule being used:  There is at least one x_1 for which`
`x_1 is a lawyer.`

**Output:** `Yes, that is one of the premises.`

This is an example, where the question is unrelated to the premises:

**Input:** `Consider the following premises.  It is not overcast.  It is overcast if and`
`only if Joseph is happy.  Can we infer the following from them?  If we can, name the`
`inference rule being used:  Karen is not an electrician.`

**Output:** `No, we cannot infer that from the premises.`

And finally, this is an example where the premises contain a contradiction:

**Input:** `Consider the following premises.  For all x_4, x_4 is an electrician.  Joseph`
`is not an electrician.  Can we infer the following from them?  If we can, name the`
`inference rule being used:  Joseph is not making tea.`

**Output:** `Yes, the premises are contradictory, so we can infer anything from them.`

### C    Appendix: Dataset Generation

This section presents some additional details on dataset generation. Specifically, we provide some additional details on how inference problems are created, and how clauses are translated to natural language.

## C.1 INFERENCE PROBLEM GENERATION

As mentioned in the paper, an *inference problem* is defined as a tuple $p = (P, I, C, U)$, with a set of premises, inferences, contradictions and unrelated clauses. Inference problems are the starting point from which training examples are generated.

To generate an inference problem, we basically create random inference chains by chaining some of the defined inference rules above. Specifically, we use the following procedure:

1. One inference rule $r_0$ is selected at random.

2. From $r_0$, we can form an inference problem $p$ as follows: the premises of the problem are the same as those in the rule: $p.P = r_0.P$. For each inference in the rule, we construct an inference chain of length one in the problem: $p.I = \{(c, [(r_0.P, c, r_0.name)]) | c \in r_0.I\}$ (each inference chain consists of the final conclusion, and a list of triples: (premises, conclusion, inference rule name) with the inference steps). We do the same with the contradictions: $p.C = \{(c, [(r_0.P, c, r_0)]) | c \in r_0.C\}$. The set of unrelated clauses are also taken directly from the rule: $p.U = r_0.U$.

3. At this point, we can now take any of the premises $c \in p.P$, and choose a random rule $r_1 \in \mathbb{R}$ that could be used to infer $c$. Then, we remove $c$ from $p.P$, and add the premises of $r_1$ to $p.P$. We then extend all the inference chains in $p.I$ and $p.C$ by adding one initial step using $r_1$ All variables in $r_1$ are renamed to unique names before using it to update the inference problem, to prevent any name clashes. This step 3 can be iterated as many times as desired to create arbitrarily long inference chains.

4. Finally, as the previous process just creates arbitrary inference chains, it is possible that the generated chains are redundant (inferring things that are already known), or even introducing contradictions. Thus, a final step to simplify inference chains (any inference step that infers something that was already known is removed, and if the statement that wants to be proven or contradicted is inferred earlier, then the inference chain is cut at that point). As mentioned, inference problems might be created in a way that their premises lead to contradictions. If any such problem is generated, it is stored on a separate buffer, and at the end of inference problem generation, some of those problems with contradictions (up to a user specified proportion) are added back to the set of inference problems, so that we have, up to the desired fraction of problems with contradictions (by default, we limit to at most 10% of inference problems containing contradictions, but in reality many less than that are generated.

The process that generates an inference problem receives a hand-defined probability distribution over the possible problem lengths (number of times we iterate step 3 above) that we want, and this is used for generating inference problems. By default, we use the following distribution: [0.425, 0.3, 0.2, 0.05, 0.025] (for 0, 1, 2, 3, 4 and 5 applications of step 3 respectively).

## C.2 TRANSLATION TO NATURAL LANGUAGE

Translation of a clause to natural language follows a set of patterns:

- Atoms of the form `p`, `q`, etc. get translated to one of these forms:
  - "subject verb-action" (e.g., `Mary plays tennis`),
  - "Subject predicate" (e.g., `Mary is happy`),
  - "Impersonal-action" (e.g., `It is raining`)
- There is a set of predefined subjects, verb-actions, predicates, and impersonal actions and they are sampled randomly (but without replacement within the same training example, to prevent repetitions).
- When an atom is of the form `P(c)`, `Q(c)`, etc. then only the patterns with subjects above are used and `c` is mapped to the subject, and `P`/`Q` to the verb-action/predicate.
- When an atom is of the form `P(x)`, `Q(x)`, etc. then only the patterns with subjects above are used and the subject is just rendered as `x` (since `x` is a variable here). We acknowledge that this step could use better translation to natural language.

- Each atom can be rendered in several modes (present, past, negated, etc.) to be used in the patterns below.
- Or, and, implication and biconditional, have patterns like `X or Y`, `if X then Y`, etc.
- Quantified clauses also have patterns: `For all x, X` and `There is at least one x for which X`.
- Finally, there is a special case for existentially quantified rules of the form `exists x, P(x) and Q(x)` that renders it as `some Xs are Y` (where `X` and `Y` are the predicates associated with `P` and `Q` respectively).

In the generation script there are currently 20 possible subjects (the 10 most common male and 10 most common female names in English), 30 possible predicates, 15 possible actions, and 8 possible impersonal-actions. For example, `p -> q`, could be translated to `If John plays Tennis then it will snow`.

## C.3 Additional Considerations

During the development of LOGICINFERENCE, we observed that models obtained much better performance in the first versions of the dataset, which was surprising to us. We realized that the generation pipeline had certain biases which models were latching onto for predictions. For example:

- Due to the way problems are generated, variable names were indicative of the particular inference chain that had used. Each time a rule is applied for generation, there is an internal counter (that starts at 1), and all variables in a rule are then renamed based on that counter (`p` becomes `p_1`, etc.). We noticed that models were able to identify patterns based on these numbers, and that's why we introduced the variable renaming variations. This made the problem harder, as removed these types of clues.
- In addition to variable renaming variations, we had to randomly shuffle the premises of a problem each time a new example was being generated, as the order in which the premises appear in a problem also contained clues that models were latching onto.
- For problems of type 1, there are many ways in which a problem can be translated to semi-formal notation. For example, if a natural language statement, can be translated to `p -> q`, then it could also be translated to `r -> s`, as it's just a matter of which names do we decide to use to assign to each predicate. This was making evaluation complex, as models were proposing translations that were correct, but different from that present in the dataset. To solve this problem, we defined a *canonical naming* convention, where the first atomic proposition that appears is always called `p`, the second always called `q`, etc. This was used only for problems of type 1.

## D Appendix: Dataset Statistics

The dataset generation script can generate datsets or arbitrary sizes. In this section, we report statistics using the default configuration, which is:

- Inference chain distribution: $[0.425, 0.3, 0.2, 0.05, 0.025]$.
- Try to generate up to 5000 inference problems.
- Try to generate 25 renaming variations.
- Try to generate 200000 training examples.
- For IID/OOD split train/test with a 0.9/0.1 ratio.
- For the length split, use problems with 4 premises or less as train, and 5 or more as test.

With these parameters, the resulting dataset (using random seed 0), has the following statistics (notice that all the splits have less than 200000 training examples, due to duplication removal):

- IID: 177543 / 19728 instances for train / test.
- OOD: 177578 / 19743 instances for train / test

- length: 175179 / 22092 instances for train / test.

More detailed statistics for one of the splits (IID):

- The script generated 4814 different inference problems generated (168 with contradictions).
- In these 4814 inference problems, we find (note that these do not add up to 4814, since each inference problem might have more than one inference chain, and also, we are counting chains in both the inferences and contradictions set):
  - 1079 inference chains of length 0 (appear in premises).
  - 607 inference chains of length 1 (application of a single inference rule).
  - 1347 inference chains of length 2.
  - 4227 inference chains of length 3.
  - 2127 inference chains of length 4.
  - 852 inference chains of length 5.
- 77199 renaming variations were generated.
- The example distribution among the different problem types was (notice that the low count of problems of type 2a is because, as they do not depend on inference chains, but only on premises, many were removed due to duplication removal):
  - 1: 24019
  - 2a: 48422
  - 2b: 37794
  - 3a: 49713
  - 3b: 37323
- Once tokenized using the T5 default vocabulary, we see the following token length distribution in the training set (min / median / 90% percentile  max):
  - Input: 18 / 97 / 147 / 244
  - Output: 2 / 70 / 269 / 582

Notice that we use input/output sequence lengths of 256 and 512 respectively for our experiments, so, the longest examples will get cropped. This only happens for less than 100 examples in the IID set, so we decided not to increase the sequence length beyond 256/512 for efficiency.

Moreover, concerning the tokenization, we also would like to note that the characters "<" and "~" are not part of the T5 vocab, and hence are parsed as "unknown". Moreover, since the "<" always appears before a "–", and "~" can never appear before a "–", we can perform a simple string replacement of unknown tokens by these two tokens easily. However, we note that this might cause some small problems for T5, and hence, an extended vocab with these two tokens might obtain slightly better results.

## E  APPENDIX: DETAILED RESULTS

**Learning Curves.** Figures 1 and 2 show the learning curves corresponding to the results in Table 2. As the curves show, evaluation sequence accuracy still continued to grow at the end of training (except for the length split when training from scratch, which had stabilized). Hence, training for longer would result in better results. Also, although we do not show it, training set accuracy reaches above 0.99 very quickly (even when training from scratch), usually after 20k iterations when training from scratch. However, as noted by Csordás et al. (2021), test accuracy continues to grow in compositional generalization tasks even after training set accuracy (or even IID validation set accuracy) has saturated.

**Accuracy by Problem Type.** It is also interesting to see in which types of problems do errors occur. Table 3, shows the performance divided by problem type in the LOGICINFERENCE$_e$ version. We report results for the base model, which generally achieved good results. Table 3 reveals an interesting effect: when training from scratch, models significantly struggle in task 1 (0.075 accuracy). Moreover, task 3b seems to be the harder task overall in the IID and OOD splits.

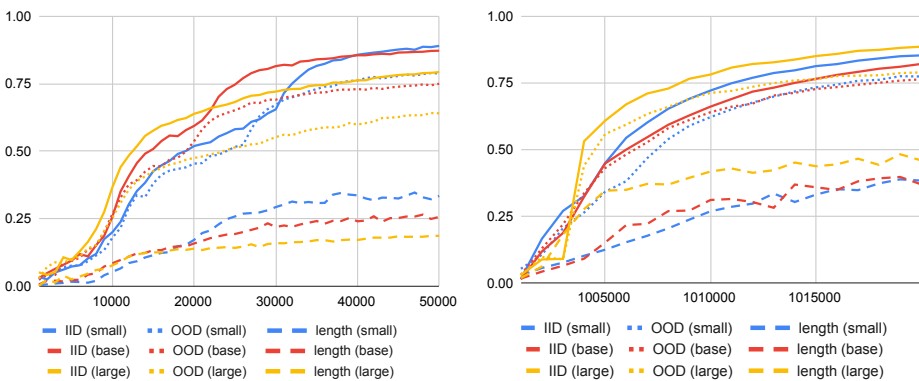

Figure 1: Training curves for the LOGICINFERENCE$_b$ (answer at the **beginning**) version of the dataset. Vertical axis is sequence level accuracy on the test set, and horizontal axis is training step. **left)** training from scratch for 50k steps. **right)** fine-tuning for 20k steps starting from the public T5.1.1 pre-trained checkpoints (pre-trained for 1m steps).

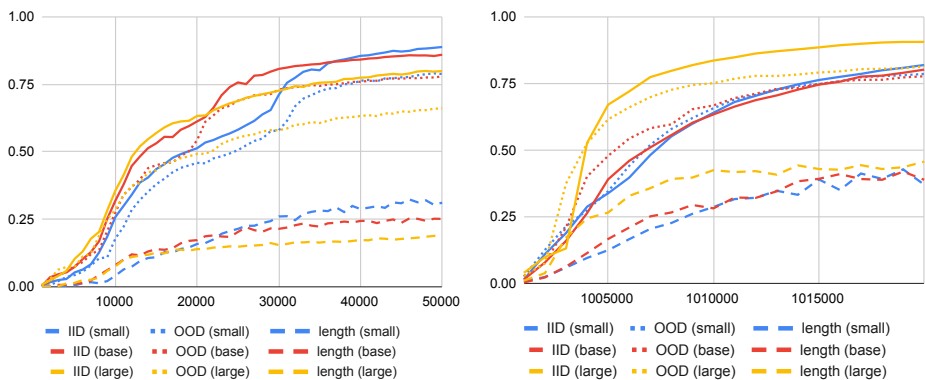

Figure 2: Training curves for the LOGICINFERENCE$_e$ (answer at the **end**) version of the dataset. Vertical axis is sequence level accuracy on the test set, and horizontal axis is training step. **left)** training from scratch for 50k steps. **right)** fine-tuning for 20k steps starting from the public T5.1.1 pre-trained checkpoints (pre-trained for 1m steps).

**Accuracy by Problem Length.** Figure 3 plots the accuracy as a function of the problem input length (in tokens). As the figure shows, examples with shorter inputs are easier to predict (higher accuracy), and errors are concentrated in the longer examples. This shows that models struggle to generalize via *productivity*. Moreover, notice that in the length split, since examples are split by the

| Problem Type | From Scratch (50k) | | | Pre-trained (20k) | | |
|---|---|---|---|---|---|---|
| | *IID* | *OOD* | *length* | *IID* | *OOD* | *length* |
| 1 | 0.987 | 0.973 | 0.075 | 0.975 | 0.980 | 0.393 |
| 2a | 0.933 | 0.884 | 0.277 | 0.888 | 0.877 | 0.416 |
| 2b | 0.885 | 0.877 | 0.255 | 0.842 | 0.854 | 0.395 |
| 3a | 0.839 | 0.673 | 0.294 | 0.722 | 0.662 | 0.396 |
| 3b | 0.678 | 0.564 | 0.255 | 0.639 | 0.595 | 0.335 |
| overall | 0.859 | 0.779 | 0.252 | 0.802 | 0.777 | 0.389 |

Table 3: Sequence-level accuracy by problem type for the base model on the LOGICINFERENCE$_e$ dataset version.

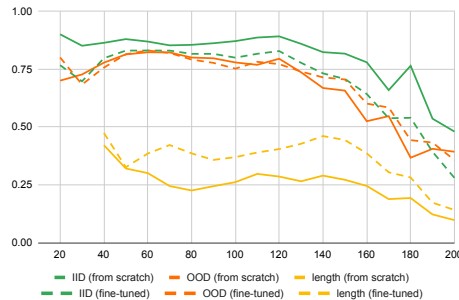

Figure 3: Sequence-level accuracy (vertical axis) as a function of the input length of the problem (in tokens, horizontal axis) for the base models in the LOGICINFERENCE$_e$ version of the dataset.

number of input premises, there are no problems with short inputs in testing, and hence, the curves start more to the right.

**Results Variance.** As we were only trying to present example baseline performance, we only report accuracy of a single run per model configuration. However, we observed a some degree of variance in the results in our experiments. In some preliminary runs with larger size models (*xl* size), we observed that while the best runs achieved better results than those in the table (0.940 accuracy in the IID split), the variance problem was more pronounced (*small* and *base* seemed to be more stable), and we decided not to include *xl* results, as that would require multiple runs and reporting averages, increasing the computation cost of the experiments significantly. Previous work on compositional generalization tasks has already reported on this type of variance in some tasks, and hence, this is not particularly surprising. However, the consequence is that future work using this dataset should probably report the average of several runs to prevent this issue.

