# OpenReview forum: "LogicInference: A new Datasaet for Teaching Logical Inference to seq2seq Models"
_ICLR.cc/2022/Workshop/OSC — ICLR2022 OSC  Poster_

### Official Review · Reviewer_WGiv · 2022-03-10
**A novel dataset for measuring performance in logical inference**

**Rating:** 2
**Confidence:** 3

**Review:**

This work presents a novel dataset, called LOGICINFERENCE,  which was developed for evaluating the compositional generalisability of AI models, specifically for evaluating the performance of models to handle tasks of logical inference. Apart from this another goal of the dataset is to evaluate how well a model trained on this dataset can transfer learned knowledge to other tasks (although this is only mentioned as a step in future work).
The dataset consists of several interesting logic subtasks presented in semi-formal logical notation as well as natural language and contains fairly reduced number of samples, making it an interesting challenge, particularly for more data hungry deep learning models. The authors provide a small baseline evaluation, focusing on different versions of the T5.1.1 model, indicating the challenge of the dataset.

In summary the work is well written and the dataset seems interesting for initiating research in more structured models for logic inference. I believe it offers interesting points of discussion for the workshop.

---

### Official Review · Reviewer_3LhZ · 2022-03-15
**Good contribution to the comminity.**

**Rating:** 2
**Confidence:** 2

**Review:**

This paper presented a new natural language dataset for evaluating compositional generalization of machine learning models in the context of logic inference. The authors proposed a natural language interface for the underlying reasoning task. The dataset is generated by synthesizing possible reasoning chains in a subset of FOL with an upper length constraint.

I believe the release of the dataset can be a good contribution to the community so I recommend accept this paper.

Following are a few weaknesses I found while reading the paper.

1. The presentation of "OOD" generalization split is unclear in the main text. I briefly skimed the appendix and couldn't find the exact setting either.
2. The authors should discuss why they chose to include only a "subset" of FOL? What are the technical/conceptual concerns?
3. This is probably the most serious concern I have. The author have compared their new dataset with existing datasets in the introduction section. For example, the dataset is a more general dataset than CLUTRR. However, there is no empirical study of how these different design choices affect our findings. For example, do models show similar generalization results on these datasets?
4. There are other mathematical reasoning datasets missing in the reference (as well as comparisons). For example, Saxton et al. "Analysing Mathematical Reasoning Abilities of Neural Models" Hong et al. "Learning by Fixing: Solving Math Word Problems with Weak Supervision"

---

### Decision · Program_Chairs · 2022-03-21

**Decision:**

Accept (Poster)

**Comment:**

The reviewers agree the paper should be accepted at the workshop. Congratulations!

The authors are encouraged to take the points raised by reviewer 3LhZ into account when preparing the camera-ready version of the paper.